# Zoofolkloristics: Imagination as a Critical Component

**DOI:** 10.3390/ani14060928

**Published:** 2024-03-17

**Authors:** Teya Brooks Pribac, Marjetka Golež Kaučič

**Affiliations:** 1European Studies, School of Languages and Cultures, Faculty of Arts and Social Sciences, The University of Sydney, Sydney, NSW 2006, Australia; 2Institute of Ethnomusicology, Scientific Research Centre of the Slovenian Academy of Sciences and Arts (ZRC SAZU), 1000 Ljubljana, Slovenia; marjetka.golez-kaucic@zrc-sazu.si

**Keywords:** zoofolkloristics, critical animal studies, animal subjectivity, animal ethics, imagination, imaginative counter-discourse, dominant narratives, counter-narratives, animals in folklore, Slovenian folklore

## Abstract

**Simple Summary:**

In this paper, we perform a critical re-reading of selected animal-related folklore texts and, applying Hubert Zapf’s concept of imaginative counter-discourse, consider the potential of imagination as a methodological tool in the transformative program of zoofolkloristics.

**Abstract:**

Nonhuman animal protagonists of folklore texts in the European space have tended to be perceived primarily as performing a symbolic and metaphoric function. But behind the symbols and the metaphors hide real flesh-and-blood nonhuman animals, and flesh-and-blood humans interacting with them, mostly from a position of power. The emerging discipline of zoofolkloristics considers nonhuman animals in their own right. Through critical analysis of folklore material, zoofolkloristics examines the role of animals and power relations within the interspecies entanglement with the aim of deconstructing the oppressive system and establishing multispecies justice. We begin this paper with a brief reflection on the ‘historical animal’ as an embodied being and a human construct. We then perform a critical re-reading of three animal-related folklore texts from the Slovenian tradition and, applying Hubert Zapf’s concept of imaginative counter-discourse, consider the potential of imagination as a methodological tool in the transformative program of zoofolkloristics. Implications for animal ethics, liberation, and conservation are also discussed.


*Religion is always right. Religion protects us against that*

*great problem which we all must face. Science is always wrong;*

*it is the very artifice of men. Science can never solve one problem*

*without raising ten more problems.*
—Bernard Shaw [1] (p. 16)

## 1. Introduction

The epigraph that opens this essay comes from a speech Shaw delivered in London, on 27 October 1930, at a dinner in honour of Albert Einstein. The speech was later utilised to preface Einstein’s book Cosmic Religion and Other Opinions & Aphorisms, which includes Einstein’s famed statement: ‘Imagination is more important than knowledge’ [2] (p. 49). Arguably, Einstein never intended unguarded imagination to supplant data collection and knowledge acquisition; rather, he understood that at any point in time knowledge is limited, and for this knowledge to expand and evolve, we need to give space to intuition and imagination. Imagination is, as he put it, ‘a real factor in scientific research’ [2] (p. 49). Shaw himself liked to push boundaries. ‘Shaw was forever unimpressed with conventional wisdom’, Rod Preece writes [3] (p. 4): everything that society takes for granted deserves a good measure of doubt and further exploration.

Other thinkers, contemporary and past, have shared this sentiment. Sixteenth-century philosopher Michel de Montaigne was among them. Knowledge—both scholarly knowledge and common knowledge—is vulnerable to habituation. Habituation may lead to inertia, and, when scholarly investigations continue within the framework of an already-formed mind, to a self-serving and self-deceiving research practice that, instead of opening up new horizons, strengthens inherited assumptions and beliefs. Foglia and Ferrari [4] (n.p.) summarised de Montaigne’s view on inertial scientific inquiry:

it makes us spend our time justifying as rational the beliefs we inherit, instead of calling into question their foundations (…) Whereas science should be a free inquiry, it consists only in gibberish discussions on how we should read Aristotle or Galen. Critical judgement is systematically silenced.[4] (n.p.)

In relation to nonhuman animals—the primary focus of the present article—this type of investigative approach has been widespread across the sciences and humanities. For example, until recently, animal consciousness and subjectivity were considered impossible to study ‘objectively’ and as a consequence were banned from the discourse. Non-observance of this doctrine imposed by the dominant narrative could cost scholars their careers and reputations [5,6], so only a few dared to imagine. The failure of the majority to openly imagine a counter-narrative enabled the consolidation of the instinctual and mechanistic view of nonhuman animal lives—a view that is progressively dissipating in the face of accumulating evidence of the psychobiological and social complexity of nonhuman animals and human–nonhuman comparability [7,8,9,10,11,12].

Habituation and imagination (or lack thereof) also inform the choice of research subjects and approaches. ‘The nature of attention one brings to bear on anything’, McGilchrist notes, ‘alters what one finds’ [13] (p. 29). For a long time, for instance, nonhuman primates were considered cognitively, emotionally, and socially superior compared to other animals. With this belief in place, scientists began developing research questions aiming at uncovering primate complexity. Sheep, on the other hand, as Vinciane Despret [14] reports, were not believed to be complex and as a consequence did not receive comparable attention from researchers. With time, evidence of primate complexity grew plentiful while evidence of sheep complexity remained non-existent, not because sheep are not complex (it turns out they are) [15], but because no one was looking for complexity in sheep. When we started to look, we found all sorts of interesting and unexpected things: for example, zebra finches dream with their tiny, half-a-gram brains, and sometimes they dream of singing [16], while octopuses turn out to be smart, curious, adventurous, and opportunistic [17].

Other disciplines are afflicted by the same habits, including folkloristics—a parallel focus of this essay. Folklore scholars study traditional behaviours, customs, and forms of art and expression preserved in human societies. Many of these works feature nonhuman animals, and, doctrinarily within this discipline, nonhuman animals tend to be interpreted symbolically, as metaphors, in some function of the human animal. Rarely, if ever, are nonhuman animals considered as real, flesh-and-blood individuals with their own lives, their own stories, needs, aspirations, relations, customs, and cultures. The folklore student’s imagination is free to run wild as long as it remains within the constraints imposed by the dominant narrative, i.e., as long as the student reads the material, as de Montaigne suggested, through the lens of the dominant paradigm. This helps to consolidate inherited conceptual frameworks, and it may hinder exploration of the multitude of views and modes of being, intrinsic to the entangled, dynamic reality of planetary existence. To redress the limitations of this traditional approach, replete with intellectual shortcomings and ethical debacles, a new branch has emerged within the discipline—namely zoofolkloristics [18,19]—which seeks to re-examine inherited traditions in light of scientific and philosophical advances in relation to nonhuman animals.

The subject of zoofolkloristics is nonhuman animals as real, thinking and feeling individuals (rather than as symbols or metaphors) in folklore, i.e., in all spheres of (human) folk spiritual culture, including songs, tales, fairy tales, fables, legends, proverbs, sayings, riddles, jokes, folk language, cultural practices, folk drama, mythology, and folk medicine. Zoofolkloristics draws on humans’ perceptions and creative representations of the world around them with a focus on nonhuman animals in that world. It is, essentially, a critical analysis of the position and role of other-than-human animals in various folklore works as well as representations of human–nonhuman relations.

Zoofolkloristics introduces theoretical and analytical approaches that enable insight into the changing attitudes of humans towards other animals in folklore as well as within traditional and contemporary ritual practices. This may facilitate a reframing of historical and current human–nonhuman entanglements with potential impact upon the legal protection of nonhuman subjects. Theoretically and methodologically, zoofolkloristics is formed in connection with ecocriticism, critical cultural and critical animal studies (CAS), literary criticism perspectives on animal issues [20], and other concepts and discourses across the sciences that help illuminate interspecies encounters, entanglements, and relations in a given (and inherently dynamic) socio-political context. Echoing the basic tenets of CAS, zoofolkloristics considers nonhuman animal issues as political. The utilisation and representation in various discourses of nonhuman animals affect real individuals and therefore carry ethical significance. At the core of CAS, we thus find an ethical reflection (with a call for action) on the relations between humans and other animals, firmly grounded in intersectionality, environmental justice, social justice politics, and a critical analysis of the fundamental role played by the capitalist system [21,22,23,24,25,26,27,28]. This critical approach also flows into zoofolkloristics, allowing the folklore scholar to consider nonhuman animals in folklore not (only) as part of ‘nature’ but as part of culture.

We begin this paper with a brief consideration of the ‘historical animal’ as an embodied being and as a disembodied, metaphorical human creation. Within this context the question of narratives arises. How reliable is the picture painted by the dominant narrative and passed down through generations? Autopoetics is insufficient, and the ensuing hierarchisation of species is progressively losing ground as multispecies, interspecies, and intersectional realities emerge, forcing us to stretch our empathic and imaginative muscles as we reconsider what it means to think of animals in a different way. However, counter-narratives have existed alongside dominant narratives: de Montaigne and Shaw are cases in point. Can a critical re-reading of folklore texts expose counter-narratives that have been missed by scholars blindsided by the dominant paradigm? To what extent can imagination help in this process? Applying Hubert Zapf’s [29] concept of imaginative counter-discourse, which was originally developed within the framework of literature and ecocriticism, to the field of folklore, in the last section of this paper we perform a critical re-reading of three animal-focused folklore texts from the Slovenian tradition and consider the utility of imagination as a methodological aid. The practical implications of such critical revisiting of traditional texts for animal welfare and conservation efforts are also considered.

## 2. Historical Animals

Over the past sixty years, through human manipulation, the size of a broiler chicken has more than quadrupled [30]. At the tender age of fifty-six days, in 1957, a chicken would have weighed 905 g. This weight would increase to 1808 g by 1978, and to 4202 g by 2005, a 459 percent increase. Other animals have also experienced a size surge, including pigs and turkeys. Body size (and the pressure this unnatural growth puts on their musculoskeletal system and internal organs) aside, neurobiologically, psychologically, and socially, animals have remained moderately stable. ‘Some changes have occurred and others will continue to occur’, Donald M. Broom and Ken G. Johnson observed, ‘but most characteristics are very resistant to change’ [31] (p. 33). Speaking of so-called domesticated species, they point out that millennia of evolution and adaptation cannot easily be outweighed by the relatively short period of domestication. This means, essentially, that what we know about various animal species today applied to those species two thousand years ago and even earlier.

The weight of evidence from wide-ranging scientific and philosophical investigations into nonhuman animal life presents a picture of nonhuman animals bursting with subjectivity and reflecting comparability to the human animal in everything that counts for survival and well-being. We are all capable of experiencing a broad spectrum of emotions, from pain to pleasure [32]. Like humans, other animals, including invertebrates, can be pessimistic or optimistic [33]. The cognitive apparatus, and the way it processes and integrates information, is also shared across species [34]. Like humans, other animals too have culture and social norms [12], traditions, and (unwritten) folklore or animalore [19,35,36] (We introduce ‘animalore’ here as an umbrella term to describe traditional beliefs and behaviours across animal species; all animal communities, not only human communities, have traditions that are passed through the generations in species-specific ways and forms of communication. Folklore is the human version of animalore). They lead creative lives [37], they experience awe, and they share with humans an a-transcendental, bodily-focused spiritual dimension with significant psychobiological implications [38]. They may even have religion [39,40].

The intense preoccupation with the brain in the past three decades, sparked by the proclamation of the 1990s as the ‘decade of the brain’ [41], has uncovered commonalities in animal brains across species, including in cases of deceiving morphology. For instance, the absence of a neocortex in avian brains has led to the mistaken assumption that birds are intellectually inferior compared to mammals (as reflected in the term ‘birdbrain’). We now know that while the avian brain is organised differently, it features equivalent structures and processes [42]. Along with commonalities, diversification is equally significant. Animals have evolved in and with different environments, foregrounding a species-specific phylogenetic normative. If circumstances pull animals too far from this normative, their quality of life may be compromised; in the event of a considerable gap between what is and what should be, life itself may be at risk. This is true now as it has been historically. A fish out of water could not survive in the Middle Ages and they cannot survive out of water today; a hen in a cage would not have thrived then and she cannot thrive now. A mother cow whose baby was taken away from her grieved in biblical times, and she grieves now. The pig facing slaughter feared the slaughterman then, and he fears him now. It is on these premises that animal historians have begun to explore and validate nonhuman animals’ experiences across temporal and geographical lines.

In the introduction to the anthology Animals as Experiencing Entities: Theories and Historical Narratives, editors Michael J. Glover and Les Mitchell note that the field of animal history, which describes the study of ‘histories with animals as prominent themes or lenses’ [43] (p. 13), has grown exponentially over the past forty years. However, until recently, most of the investigations avoided tackling the question of nonhuman animal experiences since, as intimated in the introduction, for a substantial part of recent history, animal subjectivity was not a desirable topic of inquiry. With burgeoning support from various sciences—the neurosciences, zoology, ethology, and many others—and with the growing appreciation of interdisciplinary research approaches, it is now becoming possible to reverse this situation and reinstate nonhuman animals from the past as feeling, thinking individuals with their own stories and histories.

If ‘to build an animal history, history needs the help of other sciences’, as Éric Baratay put it (reported in [43], p. 14), sciences too will at times need the help of history. Ecologist Daniel Botkin, for instance, believes that ecologists need the help of folklorists; he elaborated on this idea in a 2006 lecture ‘The Folklore behind Ecology, or Why Scientists in Ecology Need Help from Folklorists’ [44]. Reiterating de Montaigne’s concern that science can fall victim to justifying inherited false beliefs and sacrificing critical thinking, Botkin opines that folklorists can aid ecologists (and other scientists) in identifying solid traditional knowledge and help them think through how this knowledge can be used. When zoologist Andrew Whiten was asked by New Scientist’s Michael le Page whether we may be losing animal cultures before we know they exist—a form of ‘dark extinction’, as Boehm and Cronk [45] call the extinction of species before they are discovered—Whiten replied in the positive: he thinks we are. There is evidence, Whiten explained, that the degradation of habitats can lead to the shrinking of animals’ behavioural repertoire, including culturally determined behaviours [46]. It is becoming increasingly evident that cultural transmission (i.e., the horizontal, vertical, or oblique transfer of information and behavioural traditions within animal societies) plays a significant role in animals’ individual and social lives. Investigations have revealed, for example, that ungulate migrations are culturally informed. Studying bighorn sheep and moose, Jesmer and colleagues [47] observed a marked decline in migratory propensity in translocated herds compared to herds who have remained in the same home territory for an extended period of time and have had the opportunity to acquaint themselves with the area’s vital characteristics (e.g., plants’ growth cycles, predator distribution, etc.). Migratory flexibility, and behavioural flexibility generally, is clearly beneficial in the lives of all animals. A critical animalist reading of folklore and other historical accounts, proposed by zoofolklorists, could provide useful insights that have over time been lost but that could potentially significantly contribute to our understanding of species as well as of interspecies dynamics, with practical benefits for individual welfare and also for the conservation of species and their cultures.

Given that over the past few thousand years animals’ sentience and socio-psycho-biological capacities and necessities have not changed significantly, and that humans have in varying modes and degrees always been entangled with other animals, it is unsurprising that the question of how humans should treat nonhuman animals is not novel. This question has, in fact, preoccupied humans all through history. In the following section, we consider some of the approaches taken by the historical human animal.

## 3. Narratives

‘History is shaped not by weapons, or tyrants, or rebellions. History is shaped by *stories*. Beneath every oppression and every revolution are narratives that guide them’, wrote psychologist Melanie Joy [48] (n.p.). Every society has a master narrative [49]: a template that influences how we perceive the world and ourselves in it. It shapes our values and informs our actions. Alongside these dominant narratives, we find many little narratives, those *petit récits* [50], that often contradict the master narrative. In the broader Anglo-European space, the dominant narrative concerning other-than-human animals and the rest of nature has been that of division: humans have tended to consider ourselves as separate and superior to other life forms, entitled to use them as we pleased. Nevertheless, the narrative of radical separation of the human from the rest of the living world has been only one of the perspectives. Alternative, nature-based counter-narratives of connection, interdependence, human–nonhuman comparability, and trans-species communitarianism are not foreign to the European mind; they do not only belong to indigenous populations in colonised countries or to Asian traditions, as it is often depicted. They have always been part of Western cultures: for instance, the famous animal trials [51], taking place between the thirteenth and eighteenth century in Europe, were based on the assumption that nonhuman animals possess adequate levels of subjectivity and agency; another phenomenon of interest is the widespread new animism [52,53], connecting contemporary Europeans (and others) to our old pagan roots (it is worth noting, however, that paganism never truly disappeared from Europe; most customs and traditions include a mixture of paganism and Christianity). Conversely, it is not unusual—quite the opposite—to find anthropocentric traits in traditions outside the Anglo-European context [54,55].

If these counter-narratives have always existed, how could we repeatedly have missed them? Just like our scientists from the introduction to this paper focused on primate complexity and completely ignored other animal species in the belief that the latter were not complex, or Botkin’s colleagues who, according to Botkin, conducted ‘bad science’ due to their uncritical reliance on cultural influences and assumptions, humans appear to also read texts in a similar fashion. Andrew Linzey and Dan Cohn-Sherbok [56] discuss this problem in relation to the Bible. Scriptural texts are read, they note, ‘in ways that most comply with preconceived ideas and established practice’ [56] (p. 18). This results in Biblical exegesis propagating ‘an instrumentalist reading’ of Jewish and Christian sacred texts, obscuring the presence of other motifs within those texts [56] (p. 17). Linzey and Cohn-Sherbok are aware that with their focus on the pro-animal sentiment and ethics, they could be accused of engaging in yet another selective reading of the Scriptures. In reality, they identify other equally authentic elements in those texts. They are not pretending that their reading is the whole story, they are simply showing that the dominant narrative is not the whole story [56] (p. 18).

Instrumentalist and figurative readings of textual nonhuman animals, along with the culturally informed ignoring of topics, extend beyond biblical exegesis, contributing to the invisibility of nonhuman animals as sentient beings and related (animal) ethics. Erica Fudge notes that numerous early modern texts (1500–1800 CE) include accounts of animals meant to be read literally, but our cultural conditioning is so strong that even in posthumanist readings (which supposedly seek to overcome traditional and anthropocentric binaries), animals tend to be either ignored or read figuratively [57,58]. ‘Those early modern texts … seem to have much to say about beings other than humans’, Fudge writes, and when in those texts ‘we recognize the literal meaning of animals … what emerges is a vast body of literature that is in fact concerned not only with humans but also with [other] animals’ [57] (p. 10). Ancient philosophers also wrote extensively about nonhuman animals and interspecies relational ethics, but for the most part, their readers have silenced them too. ‘[T]he academic study of antiquity in the west has largely ignored early thinkers and arguments concerned with animal ethics’, Joshua J. Sias points out in his article that tracks ancient arguments for the ethical consideration of nonhuman animals [59] (p. 1). Among these early advocates, who also tended to abstain from consuming animal flesh and were critical of hunting and slaughter generally, we find Pythagoras—until the nineteenth century, the term ‘Pythagorean diet’ remained in use to denote what is now known as the ‘vegetarian diet’—Plutarch, Plotinus, and Porphyry. Over the following centuries, individuals and communities continued and further developed this line of thought (and practice), including the aforementioned de Montaigne and Shaw, and many other historical names. These are familiar to us for their achievements in other fields, less so for their advocacy for interspecies justice. Several books document this ‘longest struggle’ [animal advocacy], to borrow from the title of one of them [60]. Others include, for instance, Williams’ *The Ethics of Diet* (1883), Spencer’s *The Heretic’s Feast* (1993), Crook’s (2008) *Vegetarianism in Australia 1788 to 1948*, and the recent extensively annotated sourcebook covering the history of vegetarianism from 1430 BCE to 1969 compiled by Shurtleff and Aoyagi (2022).

As investigators continue to engage in critical readings of past texts, assumptions become challenged and a landscape of the (human) mind and ‘heart’ starts to emerge that is far more picturesque than previously imagined. For instance, due to its heavily Catholic and creationist tradition, accompanied by the reality of a predominantly peasant society, attached to tradition and facing economic hardship, the Italian society has not been perceived as particularly preoccupied with animal welfare and ethics [61] (p. 2). However, closer readings have led scholars to argue convincingly for an anti-Cartesian dimension of Italian philosophy. In the Italian tradition, Cimatti and Salzani explain, ‘a neat boundary between human and nonhuman, consciousness and unconsciousness, *res cogitans* and *res extensa* never existed’ [61] (p. 8–9). Utilising Francis d’Assisi as one of the examples, the authors note that when Francis spoke to birds, he ‘implicitly assumed that the difference between their life and his own life was not metaphysical; bird life is simply different from human life’ [61] (p. 9). Upon closer inspection, boundaries in other traditions may turn out to be more fluid than inherited knowledge would have it, perhaps validating Karl Steel’s position: as he makes an interesting case for medieval posthumanism, Steel argues against the temporal limitations of ‘post’ and ‘proto’, proposing instead that ‘posthumanism does not follow humanism: rather, it is inherent in its own claims’ [62] (p. 3).

Along with works of religion, philosophy, and literature, folklore also informs behaviours and perceptions. Can a critical re-reading of folklore texts also contribute to the rewriting of narratives? In the following section, we examine representations of nonhuman animals and animal ethics in three works from Slovenian folklore. Like Italy, historically, Slovenian society was predominantly peasant, Catholic, and harbouring a deep attachment to tradition. Simultaneously, its geographical position at the intersection of cultures and trade routes has offered opportunity aplenty for contact with other traditions and resulting multi-directional influence. The most pronounced influences upon the Slovenian tradition come from the neighbouring Germanic, Romance, South Slavic, and Hungarian peoples, but influences from remote lands are also present, including Spain (a popular pilgrimage destination), the Middle East, India, and others [63]. It is also worth noting that Catholicism appears to recognise nonhuman animals as sentient beings. For instance, the paragraph 2457 of The Catechism of the Catholic Church states that humans should show kindness to other animals. However, the Catholic church also supports and promotes instrumentalisation of nonhuman animals and humans’ right to use them for human purposes [64].

## 4. Imagination

Folklore texts give us an insight, across time and place, into human–nonhuman animal relations and human perceptions and valuations of nonhuman animals’ lives. They enable us to enter nonhuman animals’ worlds through the human lens; they expose changing human attitudes towards other animals but also some deeply rooted views that even new data seem unable to disrupt. Zoofolkloristics introduces into folklore studies the so-called ‘animal turn’ [65] as it investigates whether and/or to what extent the human–nonhuman binary can be destabilised. In folklore imaginings, we can observe the passage from a poetics of species to a poetics of the individual, and this personalised relation towards other animals enables folklore to function as what Hubert Zapf termed ‘imaginative counter-discourse’ [66,67].

One of the present authors (MGK) is developing Zapf’s triadic model of literature as cultural ecology, originally proposed within ecopoetics, for application in (zoo)folklore studies. Zapf sees literature as a powerful deconstructive and reconstructive ecological force within (human) culture whose transformational function stems from a combination of three discursive modes, namely a cultural–critical metadiscourse, an imaginative counter-discourse, and a reintegrative interdiscourse (summarised in [29]).

The *cultural–critical metadiscourse* exposes traumatogenic hegemonic and coercive structures and practices of dominant systems. Referring to the intrahuman context, Zapf cites racial and gender prejudices as examples. The oppressive system that zoofolklorists attempt to deconstruct through this metadiscourse is speciesism, i.e., discrimination based on animals’ species.

In the *imaginative counter-discourse*, the marginalised take centre-stage: they transgress the limitations the dominant society imposes upon them, sparking a narrative process with transformative powers. In zoofolkloristics, these marginalised peoples are nonhuman animals who become (symbolically) empowered through accounts that recognise their agentic and sentient nature.

Finally, the *reintegrative interdiscourse* brings together the dominant narrative of the hegemonic system and the counter-narrative of the marginalised. The latter narrative disturbs the master narrative of the extant system, creating a tension that opens doors for societal transformation. For zoofolkloristics, such societal renewal would manifest as the abolition of species hierarchisation and the establishment of multispecies justice, i.e., a theory and praxis of justice that take into consideration not only human interests but also the interests of other-than-human animals (and planetary life more broadly).

In folklore, we find traces of nonhuman animals’ real lives as well as representations of fantasy (that may be desirable). Instances of anthropomorphising may be purely metaphoric transpositions of the human into a nonhuman character or they may be realities as they appear to the creator(s) of the folklore work. These realities may be illusionary or they may be post-realities, i.e., perceptions following an experience with a specific animal or species. Frans de Waal developed the concept of ‘animalcentric anthropomorphism’ [68], which is essentially a critical way of employing anthropomorphism in our thinking and representation of other animals. It recognises that a complete understanding of another animal (nonhuman or human) is not possible. However, in a world of inter- and trans-species entanglements and relationalities, the bridging of gaps is necessary, and given the commonalities among animal species (as discussed earlier), the utilisation of human concepts in discussing other animals’ experiences and manifestations is both useful and legitimate (see also [69,70]). Michael Saward discussed this as follows:

Animals can be engaged with, looked for, traced, understood, and appreciated in new ways by humans opening up themselves to new ways of ‘reading’ and ‘writing’ them (…) But to do this is to tap into new ideas of what it means to represent, and to make representations, in the senses of both what it can involve and who can do it.[71] (p. 115)

Steve Baker believes that the way we understand an animal is inherently linked to our knowledge of the cultural representation of this animal [72]. While this is something that critical animal studies attempt to redress, for a large number of humans, it does remain a factor even today. Aspects of the context from which a text portraying a particular animal emerges are therefore noteworthy. As we introduce new concepts into the reading of folklore texts, we open up possibilities for a different representation and interpretation of folklore.

### 4.1. Animals Bury the Hunter

This framework provides conceptual foundations for the analysis of the folk song ‘Animals Bury the Hunter’ (also known as ‘Beasts Bury the Hunter’ and ‘Hunter’s Funeral’). This song is classified as a humorous ballad. It tells the story of a hunter’s funeral procession. The attendees are not the hunter’s fellow humans; rather, the procession consists entirely of nonhuman forest animals (bears, foxes, wolves, rabbits, cranes, partridges, songbirds, and others), i.e., species that the hunter would normally try to kill. The hunter is killed invariably by a bear or a wolf, and the funeral procession begins. The animals carry the dead hunter to his grave, clearly enjoying the occasion.

The procession mirrors the local Catholic funeral tradition: the procession is led by someone carrying the crucifix, followed by the priest, the coffin (in the folk song the coffin is absent and the body lies on a wooden stretcher), and the mourners (family and other people). Nonhuman animals take on these human roles. The descriptions of the nonhuman animals are rather meagre but their actions are explicit, and the role attributed to each species is also significant: the rabbits, for example, bounce around and bury the hunter. The bear is heading the procession carrying the crucifix, and eventually, the bear also censes the hunter’s body; the fox is praying as a form of mockery; the wolves howl in misery because they missed the fun of the funeral; and finally, the birds take the hunter’s soul to purgatory.

Two anthropocentric interpretations and one inclusive interpretation of the thematic and historical background of this folk song emerge from a close reading of the song that can help explain the meaning and intention behind it:(a)According to one view, the song belongs to the literary cycle of *mundus inversus* (‘the world turned upside down’) [73]: nonhuman animals take on human behaviours and inter-relations in an ironic symbolisation or metaphorisation of relations within the human species—specifically, in this case, the relation between the feudal master and the serfs. Essentially, the song functions as a disguised critique of (human) social relations.(b)According to a second view [74], the song is a critique and juxtaposition of two separate but hierarchically comparable classes within the (human) society: in this case, for instance, farmers may be mocking the hunters who work for the feudal master and belong to a different social stratum.(c)A third view, in line with the tenets of zoofolkloristics, describes the song as a representation of a time and place, or perhaps simply a frame of mind, in which humans do not consider nonhuman animals as mere automata, devoid of intrinsic value, ‘objects’ towards whom humans have no moral obligations (e.g., regarding the equality of human and nonhuman animals before the law and God, as reflected in a 1587 animal trial [75]). The ballad is not a depiction of a world turned upside down; instead, it belongs to the category of (nonhuman) animal resistance [76,77]: the animals show agency when they decide to kill and bury the hunter, and with the funeral ritual, they say goodbye to the hated and feared oppressor and sigh with relief. The transfer of a human ritual into the nonhuman world shows that nonhuman lives also matter and perhaps even that the nonhuman animal world is a lot more complex than most humans recognise. The song could be a reflection of a guilty conscience and a manifestation of the capacity for empathic identification with our animal kin. Ironisation with inversion of the human into the nonhuman seems to often have been the only way of representing a departure from normativity [78,79]. If the author’s or authors’ only purpose was to expose the problem of the killing of free-living animals and the fear and subsequent relief of these animals, the goal has been achieved.

The Slovenian ballad tradition contains a whole cycle of songs featuring nonhuman animals taking on human characteristics and behaviours. Similar themes also appear on the famous beehive panels (a form of folk art that emerged in the eighteenth century and consisted of painting beehives). These motifs include nonhuman animals’ weddings, bears shooting hunters, bears chasing hunters out of the forest, and others. It is unclear how the story of the hunter’s funeral procession entered the Slovenian tradition. It currently exists both as a ballad (in thirty-one variants) and as a painted beehive panel. The motif most likely originated in the fourteenth century French collection of folk stories Roman de Renart [78,80] and then spread across Europe, mostly through images (e.g., lithographs and handbills), but not orally. In Hungary and Germany, for example, the motif is widespread but not in the form of a narrative. Even though the exact time frame is difficult to establish because the Slovenian folk tradition was not recorded in written form until the nineteenth century, there is good reason to believe that the Slovenian ballad with this motif had existed before the beehive panels were painted (the first panel is dated 1787), as well as before the lithographs with this motif reached Slovenia.

The ballad first appeared in written form before 1873 in the western part of the country; all other variants date between 1894 and 1999 (those from 1960 onward are sound recordings), and they emerged in different parts of the country (east, northwest, and southeast). The core is the same in all variants; they only differ in details. For instance, while in most variants the wolves are sad because they have missed the funeral, in some (such as the first recorded variant), all animals, including the wolves, are present and part of the procession. Of all the variants of the ballad, only one (a relatively late one, from 1960; GNI M 23.527) mentions a dog. The dog is the hunter’s faithful companion who mourns the hunter. On the beehive paintings, which include the dog, the dog is the only land animal walking on four legs, all of the others are standing upright, walking on their hind legs (Figure 1).

Clearly, from the viewpoint of the hunted forest animals, the dog is an enemy: the dog helps the hunter identify his victims and is therefore viewed to be part of the oppressive system. From a dog’s perspective, however, reality may appear more nuanced: domestication rarely brings bidirectional benefits, as we will see in the tale that follows, and in the case of working animals, life is measured by productivity: when the latter decreases, the worker/servant is often discarded.

### 4.2. About a Doggie

With the folk tale ‘About a Doggie’ (‘Od pesića’; ATU (Aarne–Thompson–Uther Index, an international index of folktale types) 20D), we move from the forest with its rebellious free-living animals to a (human) homestead with its rebellious domesticated animals. The tale originated in Rezija (Resia in Italian) on the west, near and across the Italian border. The Resian tradition stands out as it is infused with animistic traits: all beings (other animals, plants, mountains, etc.) are viewed as being organically connected with the human inhabitants [81]. For instance, the souls of the dead humans live on in snakes, and therefore the killing of snakes in that area is culturally forbidden.

Bruce Thomas Boehrer notes that early modern animal fables depicting nonhuman animal characters ‘openly create a space for the interaction of human and nonhuman species, allowing for [other-than-human] animal and human wisdom to overlap’ [82] (p. 17). Canine wisdom is emphasised in this fable, which begins as follows:

Once upon a time there was a little doggie, they tortured him at the homestead, they broke his leggie. He worked so hard that he broke the chain, you see, he broke the chain and said that he was going abroad, that he would never come back to this homestead.(Archive of the Institute of Ethnology ZRC SAZU, Archive ISN, R10/2, T 205 A 2 (2), 82–169. Translated by the present authors.)

The dog set off, and on his way, he noticed some children stretching a cat violently. They were going to nail her onto a crucifix ‘like God’. They had the wood and a hammer. The dog barked (‘woof-woof’) and threatened to bite the children. The children escaped, leaving the tools behind. The dog realised that the cat was not loved by her humans so he suggested she came with him. He was going to Rome (considered a place of plenitude as well as a popular pilgrimage destination), and as he cleverly pointed out, if they did not crucify her that day, they would do it another day. They went on and came across a rooster. They told the rooster that his (human) mistress would fatten and bake him, so they invited him along. Then, they met a goat on the way who was headbutting a tree out of misery because the humans had killed her two kids. They all walked off together and they met a donkey whom the (human) master had tied up and was going to kill. The donkey joined them, too. They all went to a house together where they drank and ate, and made themselves comfortable, each in their own corner. Then, the wolf came; the other animals attacked, killed, and ate the wolf; and that house ended up being their ‘Rome’, i.e., a safe place (cf. Brothers Grimm’s tale ‘Town musicians of Bremen’).

As in the previously addressed funeral ballad, in this folk tale, we also witness an act of rebellion and a rejection of human dominion. The dog, chained and abused, decides to set himself free and inspires other oppressed animals to follow suit. This work utilises human language, a principal ingredient of anthropomorphism, but it is in the function of communicating nonhuman animals’ subjectivity, critiquing species hierarchisation and its inherent potential for abuse of power, and enabling nonhuman animals to express their agency [83]. Implicitly, in this folk tale, human violence against other animals is denounced and the right of domination is questioned. The narrative establishes nonhuman animals’ subjectivity at the very beginning: the rebellion against human dominance builds a subject who then acts independently both as an individual and community member throughout the narration.

This folk tale could be interpreted in an anthropocentrically metaphorical way, i.e., nonhuman animals acting as substitutes for human victims of socio-economic stratification. However, it would be hard to deny that in order to create ‘even just’ a metaphorical construct, the author(s) would have had to recognise the suffering and injustices imposed upon the nonhuman species utilised (a similar case can be made for the anthropocentric interpretations of the hunter’s funeral). Additionally, and importantly from a zoofolkloristic perspective, while the animals in the folk tale may be fictional, they are also representations of real animals, past and present, who have suffered similar injustices at the hands of humans: humans do torture dogs and cats; humans do abduct and kill goats’ children; humans do kill and bake roosters; humans do abuse donkeys; and so on. An element of surprise, from an interspecies justice point of view, appears at the end of the story when these previously mistreated animals attack and kill the wolf. However, from the viewpoint of the animals involved it is understandable, since not only was the wolf an ‘other’ in that he did not belong to an enslaved, domesticated species, but had they not killed him, he may have killed and eaten them, or at least some of them. Additionally, it is not unusual for humans (including perhaps human authors of folklore works) to view predator species as evil and not worthy of moral consideration [84,85,86,87].

### 4.3. About a Pig

With the next folk song, ‘About a Pig’ (‘Od prašička’), we focus on one animal and a specific context of slaughter: the pig and the traditional cultural practice of *koline* that takes place in the period before Christmas, specifically in the first week of Advent. *Koline* refers to the act of killing and preparing, in various ways, the body of a slaughtered pig for human consumption, but it is also used as a synonym for pig’s ‘meat’. A calendar of unknown origin from the fifteenth century includes the earliest known representation (in visual form) of *koline* in the Slovenian territory [88]. The song ‘About a Pig’ appeared sometime between the fifteenth and nineteenth centuries, but mostly likely in the early nineteenth century. It features traits of rebellion, compassion, empathy, denial of all of these, and coping mechanisms inclusive of alcohol and humour.

The song tells of a (presumably) young male servant who was chosen to go into the barn and bring out the pig for slaughter. The young man tries to resist:
*For a while I was thinking about it,**And trying to get out of it.**Because I’m too fearful,**Not vigorous enough.*

Then, they gave him schnapps to make it easier. He went into the barn:
*This is what I want to tell you:**The way the pig looked at me!**When I came into the barn**He thought I was bringing him food.*(Archive of the Institute of Ethnomusicology ZRC SAZU, Arch. No. GNI O 10.616. Translated by the present authors.)

From the pig’s viewpoint, this was an act of betrayal of great magnitude, and the young man clearly realises this. The pig pricks his ears and listens, as the young man’s ‘soul trembles’. The young man tries to catch the pig, but the pig escapes and everyone else goes after him while the young man, clearly still shaken, remains in a corner. The other humans scold him afterwards, but he protests that he had told them not to send *him*. The song ends with the young man complaining that his suit had been ruined in the process. This helps to dilute the emotions and reinforce the perception of normality of *koline* and the killing of nonhuman animals generally, because no killing means no meat.

Carol Adams notes:

Through butchering, animals become absent referents. Animals in name and body are made absent as *animals* for meat to exist. Animals’ lives precede and enable the existence of meat. If animals are alive they cannot be meat. Thus a dead body replaces the live animal. Without animals there would be no meat eating, yet they are absent from the act of eating meat because they have been transformed into food.[89] (pp. 20–21)

The construction of animals as non-existent in the act of dying and the transformation from a cadaver into meat has been around for a long time and it tends to be perceived as ‘normal’. It is thematised in folklore, visual arts, literature, and religion [90]. In reality, this ‘normal’ act is a violent closure to an animal’s life. This violence is regularly overlooked by scholars analysing such thematisations, even when the latter include clear signs of the perpetrator’s discomfort and sometimes even traits of entire counter-narratives. Niko Kuret, for example, in his book Slovenians’ Festive Year (orig. Praznično leto Slovencev, 1989) refers to *koline* as a ‘festival’ and ‘a time of joy’ [91]. Kuret’s focus is anthropocentric, wherein *koline* is a happy event because it provides humans with food for the festive Christmas period. The expectation of the birth of one animal (Jesus, human), which Advent symbolises, is accompanied by the torturous death of another animal (pig, other-than-human). According to traditional folklore scholarship, humans work hard and feed the pig over the year, so they ‘deserve’ to eat the pig; after all, in accordance with ‘God’s design’, the pig’s entire existence is in the function of becoming ‘meat’ for human consumption. Kuret—unlike the song itself—completely ignores the frightened sentient being who is at the centre of it and does not want to die. Even when Kuret discusses manifestations of pigs’ agency, there is no compassion for the pig, only concern for humans. One of the present authors (MGK) has suggested that what we may be witnessing is actually a deconstruction of compassion as a sentiment of the ‘female’ (not necessarily specific to biological genders) and hence of weakness [19], a typical leitmotif of patriarchal science [92]. Even though in the song ‘About a Pig’ empathy is evident, traditional ethnologists insist that what we are seeing is fear rather than empathy [91,93], with fear being considered the more ‘natural’ and manly feeling because without fear there would be no bravery (and bravery is presumably a sign of masculinity).

There is little doubt that fear was involved, but fear is not the whole story. Zoofolkloristics needs to go beyond the stale, typically white male perspective to explore and expose other aspects that are present in folklore texts but have readily been suppressed by dominant ideologies. The unmistakable message of the song ‘About a Pig’ is that the pig is a sentient, intelligent being who clearly has a lot of agency, and that the humans go to great pains to get through the killing process with as little psychological damage as possible. For sure, when the pig’s screams subside, both in reality and memory, the humans will enjoy the ‘meat’, but while the screams subsist, they are loud, expressive, and penetrate deeply. The present authors have both witnessed the killing process of *koline.* One does not need to possess a developed philosophical position on the rights and wrongs of killing and species hierarchisation: the screams and the entire killing process awaken our mirror neurons and produce a potent physiological response. (Mirror neurons activate both during the execution and during the observation of a particular action. For instance, if we see someone biting into a lemon, our mirror neurons activate the way they would if we ourselves were biting into the lemon.) This is something we tend to forget now that the killing of animals for food has been removed from public view and it takes place behind the closed doors of slaughterhouses [94]. It may also easily be missed by scholars who read this material from the comfort of their armchairs, but for the humans who are (or were) there, in medias res, the pig (or other animal) is, at least for a while, very much a *subject*. This—along with the fact that in the past, the humans actually knew the pig, as they lived with the pig for months before slaughter, they fed the pig, interacted with the pig, etc.—helps to explain why humans have developed various coping mechanisms to help them through the killing process.

Kuret himself notes, for instance, that in some parts of the country, the lady of the house would prod the pig’s eyes with a birch broom, blinding the pig to their own death [91]. Sometimes, the pig would become dangerous because the pig ‘realised that their life was at stake’ [91] (p. 265). These are two examples that show humans acknowledging the pig’s subjectivity. Among the coping mechanisms used by the perpetrators, there is the belief that the slaughtered animal does not die if someone present feels sorry for that animal. If someone sympathises with the pig, the pig will not release much blood (the blood is used to make a type of sausage), the meat will be harmful, and humans themselves will die a difficult death [91] (p. 575). When a skilled killer is at work, the pig’s screams and rasps fade quickly [91] (p. 266), a desirable outcome.

Folklore songs tend not to openly express compassion for a nonhuman animal intended for slaughter because certain traditional forms of violence, such as *koline*, have become culturally accepted and woven into society’s psychological fabric. However, despite this acceptance, one may feel that the author(s) of the song ‘About a Pig’ needed to justify the act of killing, ironise it, and turn it into a humorous affair in an attempt perhaps to silence a guilty conscience. Either way, despite a strong utilitarian perspective, the conceptualisation of nonhuman animals in folklore is never simply an objectification of their deaths, i.e., a turning of a body into meat. Other motifs, ideas, and sentiments colour the complex landscape of custom and tradition; after all, the songs are not depicting a distanced, mass production of animals’ deaths. Rather, they are concerned with one death alone, and this death is happening in the here and the now, in front of our eyes, and in our ears.

Folklore texts that thematise close relations between human and nonhuman animals are problematic from the perspective of critical animal studies and anti-speciesism because these relations tend to be based on ownership (of the nonhuman animal by the human) rather than equality. Furthermore, they tend to take anthropocentrism and slaughter for granted. However, what is absent in these texts is the alienation between humans and nonhumans that is typical for contemporary perspectives; at the time of emergence of these works, humans and other animals lived closely together and the interconnection was more pronounced [95]. The virtue of compassion in the song ‘About a Pig’ is not a product of ‘abstract moral reasoning’ [96] (p. 81), but it derives from the immediate proximity of the human animal and the nonhuman animal who expects food rather than death.

Within zoopoetics lies zooethics. This song could be said to reflect what Tomaž Grušovnik [97] calls a ‘weak form of animal ethics’, i.e., an ethical stance that remains anthropocentric, but does not entirely objectify the pig: the pig is not just ‘meat’, (s)he is an other/kin, toward whom one can develop compassion [98,99,100]. The killing of the pig makes the killer uncomfortable. The pig does not allow their killer to remain indifferent. It is a physical experience. The farmer may have a conceptual dominion over the pig, but the pig has an emotional dominion over the farmer. (This also holds true for workers in slaughterhouses. Attempts at objectification of sentient subjects regularly fail, leaving the slaughterers vulnerable to psychological problems (e.g., Dillard 2008).)

## 5. Conclusions

Naama Harel invites us to read fables in a literal way [101]. Many of them were composed for that purpose [57]. Even in works in which the allegory appears dominant, the literal level continues beating throughout the body of the tale. Nonhuman animals in these works can exhibit physiological characteristics and capacities reflecting those of their living conspecifics. Importantly, seeing nonhuman animals in these stories as animals and not just figures, especially in stories depicting a human–nonhuman relational context, opens up space to consider interspecies issues and examine human treatment of other animals, Harel suggests [101]. Poirier points out the potential effectiveness of discussing oppression through a fictional lens:

Readers understand that the actual animals in the story do not exist and were not abused. However, similar animals do exist and are abused in similar ways. Different parts of a story may resonate more or less strongly with individual readers, whereas with a philosophical argument, if one premise is considered inadequate, the whole of the conclusion may be deemed invalid.[102] (p. 1)

*Homo narrans* [103] likes composing narratives as well as metanarratives, i.e., stories about stories. These stories, both primary and secondary, have tended to marginalise nonhuman animals, but they have never managed to eliminate them entirely. Even a homocentric culture cannot escape the entanglement inherent in the planetary organism in which it is rooted, and in all cultures, there have been individuals who have sought rectification of the moral conundrum in humans’ treatment of other animals. Folklore works, like other creative endeavours, include manifestations of both perspectives. Folklore scholarship, like most other disciplines, has in the past prioritised the anthropocentric perspective, contributing to the silencing and disempowerment of other animals. Zoofolkloristics operates in the opposite direction by actively looking for nonhuman animals in folklore texts and considering them in their own right.

In his project of ‘undisciplining the study of religion’, Kocku von Stuckrad calls for a critical posthumanist approach that ‘intentionally leaves behind the regimes of mastery and exploitation that are still operative today’ and ‘creates a transversal field of knowledge, consisting of human and other-than-human intra-actions’ [104] (p. 616). In a similar vein, through micro- and counter-narratives that are present or potential in folklore material, and taking into account new scientific understandings and philosophical reconsiderations, zoofolkloristics performs the ethico-critical work that has become both urgent and unavoidable as we face the collapse of ecosystems, extinction of species (with threats to the human species itself), and ubiquitous suffering of our animal kin on a scale without precedent.

A close reading of folklore texts that include nonhuman animals exposes an oppressive speciesist system and the suffering of nonhuman animals at the hands of humans. Simultaneously, it reveals that this violence gives rise to an unease in the human perpetrators themselves, which is mediated through the creation of unfounded associations (e.g., the belief that if one feels sorry for the slaughtered pig the meat will be bad), humour (which is emerging as a useful tool to mitigate pain [105]), and substance abuse (e.g., alcohol). In Zapf’s terminology, this is the cultural–critical metadiscourse that zoofolkloristics aims at deconstructing via the imaginative counter-discourse. The latter consists of placing marginalised beings (nonhuman animals) as well as marginalised emotions (human guilt, sympathy, etc.) in focus and imagining a new life for both. This can facilitate a distinct paradigmatic change and the shift from the role and significance of nonhuman animals to a redefinition of tradition. Nonhuman animals are no longer objects, they are subjects, with their own interests; they have intrinsic value and are worthy of moral consideration. In this process, imagination’s role as a critical component is two-fold: imagination is critical in the sense that it is essential to the process and in the sense that it aids the analysis and critical evaluation.

This deconstruction-cum-reconstruction of the past liberates both humans and other animals from practices (such as animal agriculture) that may have been considered necessary in the past and that are not only redundant but outright dangerous in the present. Cumulative evidence of the damaging effects of animal agriculture is becoming increasingly harder to ignore as calls for the transition to a plant-based diet grow louder and more numerous. Recently, David Attenborough reminded his audience that by ‘shift[ing] away from eating meat and dairy and mov[ing] towards a plant-based diet (…) we could still produce enough food to feed us but using a quarter of the land’ [106] (n.p.). A land surface equivalent to the size of the United States, China, the European Union, and Australia combined could be freed and regenerated. Despite the evidence, many humans resist this change, often citing a presumed ‘naturalness’ of the meat diet along with respect for tradition as reasons for such resistance. This transition may be facilitated by people learning that not only were early humans consuming predominantly plant-based foods [107], but that our own closer ancestors (from a hundred, two hundred, three hundred, etc., years ago) were embedded in a tradition that was a lot more nuanced than we have been led to believe. This tradition featured complex interspecies relations, and complicated interactions between human and nonhuman subjects, each with an interest in their own life and well-being. It is a tradition that might have made it *easier*, but never easy, to kill someone who does not want to die.

## Figures and Tables

**Figure 1 animals-14-00928-f001:**
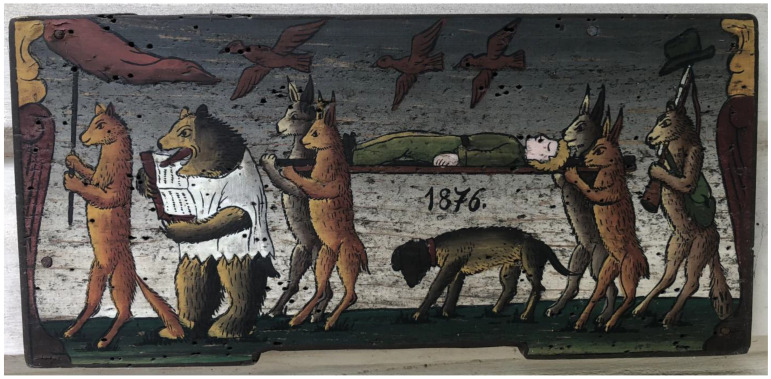
Painted beehive panel.

## Data Availability

The original data presented in the study are openly available in the Archive of the Institute of Ethnology ZRC SAZU, Slovenia, Arch. ISN, R10/2, T 205 A 2 (2), 82–169, and the Archive of the Institute of Ethnomusicology ZRC SAZU, Slovenia, Arch. No. GNI O 10.616.

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
