# Peer review of "Zoofolkloristics: Imagination as a Critical Component"

_animals, 2024, doi:10.3390/ani14060928_

Round 1
Reviewer 1 Report
Comments and Suggestions for Authors
This is a very well argued piece of scholarship that addresses a highly relevant topic. The first three sections are excellent. It is the fourth section ("Imagination", addressing the examples) that I have some questions and concerns about. I think it is important to provide concrete historical context to these stories and images, otherwise it will be difficult to interpret the data in a convincing way (instead of generalizing some hypothetical meaning of the stories). Studies into myth and fairytale have the notorious problem that they have to deal with dozens (sometimes hundreds) of variants of the story that often strongly contradict one another. Therefore, historical accuracy is important to make a convincing point. Only then, I think, can the authors also analyze the respective stories in the frame of hegemonic or counter-hegemonic discourse (the same story can be hegemonic in one context, and counter-hegemonic in another).
I also feel that the conclusion could benefit from some revision and further explanation.
As for other comments, please see the attached file. I hope the authors will find my comments useful.
Kind regards, Kocku von Stuckrad

Author Response
Thank you for taking the time to review the manuscript and provide valuable suggestions.
Dark extinction added. Thank you!
pp. 8–9: further info added. The problem with dates is that until the 19th century the Slovenian folk tradition survived principally in oral form.
pp. 10–11: The killing of the wolf is a puzzle. We did not change anything in this section because our original formulation acknowledges the inconsistency of the story, and we suggest that the folk author(s) may have been influenced by the general demonisation of predators. It is a piece of folk literature, it didn’t go through peer review :-) so we can’t expect complete consistency. On the other hand we may be missing something, but either way we do acknowledge that it is a puzzle. From our paper: “An element of surprise, from an interspecies justice point of view, appears at the end of the story when these previously mistreated animals attack and kill the wolf.” And we are talking about a wolf not a bear, a bear would probably not have killed the domestic animals but the wolf might have.
p.12 re compassion. The protagonist of the song is a male, there’s no gender issue in there. What we are saying is that ethnologists (also male and “old school”) have tended to refuse to recognise (or even discuss the possibility of) compassion in the song because that’s not something that you did in (patriarchal) science. We have added clarifying info.
p.13: terms ‘superstition’ and ‘myth’ removed from the entire text. Thank you! Concerning the lady of the house, as mentioned previously, we were not discussing actual genders or implying that women are more compassionate than men, we were exposing ethnologists’ patriarchal tendencies in their discussions.
Reviewer 2 Report
Comments and Suggestions for Authors In the abstract, it would be useful if the author mentioned the text of Naama Harel"The Animal Voice Behind the Animal Fable", which they cite in the conclusion of the paper.
If they think it's necessary.
It would also be useful, if the authors wish, to refer to the work of Kenneth Shapiro and Marion W. Copeland, “Toward a Critical Theory of Animal Issues in Fiction”, 2005, which established the goals of animal literary criticism. It seems to me that the mentioned text, as well as some other texts published in that issue of the magazine, can be perfectly applied to the key framework of this article, which is the application of H. Zapf's theory, that is, as the authors state: “Applying Hubert Zapf’s concept of imaginative counter-discourse, originally developed within the framework of literature and ecocriticism, to folklore, in this paper we perform a critical re-reading of selected folklore texts, and consider the usability of imagination as a methodological aid.“ (p. 1.). I congratulate the authors on the article.
Author Response
Thank you for taking the time to review the manuscript and provide valuable suggestions.
Reference to Shapiro and Copeland has been added.
Reviewer 3 Report
Comments and Suggestions for Authors
This article is an interesting read. There are some minor issues with content/paragraph organization, and certain claims definitely need to be supported with data. The conclusion section seems to include new sources, which might not exactly be correct, but that section seems well-rounded. Overall, I find it an original and bold piece of research that deserves to be printed after doing some minor corrections. I attach a copy of the paper with sections that might require revision underscored in yellow.

Comments on the Quality of English LanguagePunctuation and paragraph length need to be improved. Syntax should be simplified to achieve a more direct approach to content. Certain paragraphs might be reorganized to achieve a higher coherence and produce a higher impact on the readers.
Author Response
Thank you for taking the time to review the manuscript and provide valuable suggestions.
We’ve incorporated most suggestions.
Exceptions:
First comment: why should the reader care? This section established that neuropsychologically animals haven’t changed significantly through history, which means that we can discuss the historical animals’ subjectivity based on what we know about them now.
Second comment concerning the weight of evidence: all the claims are referenced using relevant works; the issues have been discussed at some length within the relevant disciplines; if the editors agree we’d prefer to keep this paragraph as is.
We’ve left the fish out of water etc. paragraph in the original position because it serves to illustrate the preceding statement and makes for a nice transition to the next section.
Reviewer 4 Report
Comments and Suggestions for Authors
Comments on the Quality of English LanguageAuthor Response
Thank you for taking the time to review the manuscript and provide valuable suggestions.
We’ve incorporated most suggestions.
Concerning Footnote #1: we do not claim that they have stories, we claim that they have traditional beliefs and bahaviours. The latter is widely accepted because of recent scientific and conceptual work in animal cultures, which is mentioned in the article.
Reviewer 5 Report
Comments and Suggestions for Authors
The article is well written and demonstrates a wide body of knowledge about the undeveloped discipline of zoofolkloristics. The literature review and history of the field is very helpful.
The suggestion of alternative readings of folklore texts, especially with Zapf's triadic model is noteworthy. However, there is little support offered for the skepticism toward the dominant narrative. Perhaps a little more attention could be given to identifying the theory and practice of the dominant narrative before it is judged to be deficient or inferior. The Catechism of the Catholic Church (par 2457) directs humans "to show [animals] kindness" and that their use must derive from a "just satisfaction" (emphasis mine).
Author Response
Thank you for taking the time to review the manuscript and provide valuable suggestions.
Added contextualisation of dominant narrative & reference to the catholic catechism